# Genetic Evidence for a Causal Relationship between Hyperlipidemia and Type 2 Diabetes in Mice

**DOI:** 10.3390/ijms23116184

**Published:** 2022-05-31

**Authors:** Lisa J. Shi, Xiwei Tang, Jiang He, Weibin Shi

**Affiliations:** 1Department of Radiology and Medical Imaging, University of Virginia, Charlottesville, VA 22908, USA; lisajshi@utexas.edu (L.J.S.); jh6qv@virginia.edu (J.H.); 2Department of Statistics, University of Virginia, Charlottesville, VA 22908, USA; xt4yj@virginia.edu; 3Department of Biochemistry & Molecular Genetics, University of Virginia, Charlottesville, VA 22908, USA

**Keywords:** dyslipidemia, type 2 diabetes, quantitative trait locus, low-density lipoprotein, high fat diet

## Abstract

Dyslipidemia is considered a risk factor for type 2 diabetes (T2D), yet studies with statins and candidate genes suggest that circulating lipids may protect against T2D development. *Apoe*-null (*Apoe^-/-^*) mouse strains develop spontaneous dyslipidemia and exhibit a wide variation in susceptibility to diet-induced T2D. We thus used *Apoe^-/-^* mice to elucidate phenotypic and genetic relationships of circulating lipids with T2D. A male F2 cohort was generated from an intercross between LP/J and BALB/cJ *Apoe^-/-^* mice and fed 12 weeks of a Western diet. Fasting, non-fasting plasma glucose, and lipid levels were measured and genotyping was performed using miniMUGA arrays. We uncovered a major QTL near 60 Mb on chromosome 15, *Nhdlq18*, which affected non-HDL cholesterol and triglyceride levels under both fasting and non-fasting states. This QTL was coincident with *Bglu20*, a QTL that modulates fasting and non-fasting glucose levels. The plasma levels of non-HDL cholesterol and triglycerides were closely correlated with the plasma glucose levels in F2 mice. *Bglu20* disappeared after adjustment for non-HDL cholesterol or triglycerides. These results demonstrate a causative role for dyslipidemia in T2D development in mice.

## 1. Introduction

Type 2 diabetes (T2D), which accounts for >90% of diabetic patients, is a heterogeneous group of disorders that share a common feature of chronic hyperglycemia resulting from defects of insulin secretion, insulin action, or both [1]. It is the most common metabolic disorder and constitutes a major health burden worldwide [2]. Despite the fact that environmental factors such as high-calorie diet and lack of exercise play a role in T2D, genetic factors are a major determinant for the development of the disorder [3]. Heritability estimates for T2D and its key components are high, with some estimates exceeding 50% [4,5]. Only a small subset of type 2 diabetic cases are caused by monogenic mutants observable as Mendelian traits segregating in families. These genes include *INSR* [6], *HNF1A*, *HNF1B*, *HNF4A* [7], *LEPR* [8], *CPE* [9], *AGPAT2* [10], *PPAR* [11], and *AKT2* [12]. The common forms of T2D involve multiple genes and exhibit significant gene–environment interactions. The latest meta-analysis of genome-wide association studies (GWAS) with Europeans identified over 100 common and a few rare variants associated with T2D [13]. However, these loci only account for a small proportion (<10%) of the heritability of T2D. Moreover, effect sizes of the loci detected by GWAS are generally small, and so identification of the underlying causal variants is challenging. Therefore, parallel approaches need to be undertaken to facilitate the identification of genes for T2D by using animal models.

Phenotypically diverse mouse strains provide a powerful experimental system for the mapping and functional analysis of genes contributing to human diseases, including T2D [14]. Almost all of the genes in mice share functions with the genes in humans, and the two species are highly comparable in development, physiology, and genome organization [15,16]. *Apoe*-null (*Apoe^-/-^*) mice are extensively used for dyslipidemia and atherosclerosis research [17]. We have found that *Apoe^-/-^* mice with certain genetic backgrounds, such as C57BL/6J and SWR/J, develop significant hyperglycemia and T2D when fed a Western diet, but become resistant after being transferred to certain other backgrounds like BALB/cJ (BALB) [18,19]. The variation among *Apoe^-/-^* mouse strains in susceptibility to T2D provides a tool to identify the genetic basis of the disorder.

Dyslipidemia, comprised of elevated plasma levels of non-HDL cholesterol and triglycerides and/or reduced levels of HDL cholesterol, is considered a risk factor for T2D. Many studies, including prospective ones, have shown that low levels of HDL cholesterol and elevated non-HDL cholesterol and triglyceride levels are associated with increased risk of developing T2D [20,21,22]. Similarly, plasma non-HDL and triglyceride levels show positive correlations with glucose levels in segregating an F2 population derived from *Apoe^-/-^* mouse strains [23,24,25]. However, several lines of evidence have suggested a protective role for circulating lipids in T2D. Lipid-lowering therapies with PCSK9-specific antibodies or statins, which inhibit 3-hydroxy-3-methyl-glutaryl-coenzyme A reductase (HMGCR), are associated with an increased risk of new-onset T2D [26,27]. Individuals with low levels of non-HDL cholesterol exhibit an increased prevalence of incident T2D [28]. Patients with heterozygous familial hypercholesterolemia caused by mutations in *LDLR*, *APOB*, *APOE,* and *PCSK9* show a reduced vulnerability to T2D compared to the general population [29,30]. Furthermore, genetic studies have revealed certain genetic loci, including *HMGCR*, *APOE*, *PCSK9*, *NPC1L1*, *PNPLA3*, *TM6SF2*, *GCKR*, and *HNF4A*, that exhibit an opposite allelic effect on lipids and diabetic traits [31,32]. Here, we sought to use a segregating F2 population from two *Apoe^-/-^* mouse strains to examine the phenotypic and genotypic relationships of plasma lipids with T2D.

## 2. Results

### 2.1. Fasting versus Non-Fasting Lipid and Glucose Levels

Non-fasting and fasting plasma lipid and glucose levels were measured after the F2 mice were fed 11 and 12 weeks of the Western diet, respectively (Figure 1). Compared with the fasting samples, the non-fasting samples obtained from the same F2 mice had modestly higher glucose levels (393 ± 132 vs. 368 ± 168 mg/dL; *p* = 0.034), slightly higher HDL cholesterol (29.8 ± 12.4 vs. 26.5 ± 14.4 mg/dL; *p* = 0.006), negligible differences in non-HDL cholesterol levels (1,178 ± 274 vs. 1,205 ± 483 mg/dL; *p* = 0.40), and modestly lower triglyceride levels (162 ± 72 vs. 186 ± 107 mg/dL; *p* = 0.0005). Mice with a fasting plasma level of ≥ 250 mg/dL were considered diabetic. Based on this criterion, 105 of 143 F2 mice (73.4%) developed diabetes, 38 (26.6%) did not, and 2 F2s had missing data.

Fasting and non-fasting glucose levels were closely correlated (r = 0.617; *p* = 2.3 E−16) (Figure 2). Correlation coefficients between fasting and non-fasting lipid levels varied from 0.480 for HDL cholesterol (*p* = 1.5E−9) to 0.606 for non-HDL cholesterol (*p* = 1.1E−15) and 0.679 for triglycerides (*p* = 1.1E−20).

### 2.2. QTL Analysis of Fasting versus Non-Fasting Plasma Lipid Levels

Genome-wide scans detected a QTL, named *Nhdlq18*, on chromosome 15 affecting fasting and non-fasting non-HDL cholesterol (Figure 3A,B) and triglyceride levels (Figure 3C,D). This QTL had a confidence interval varying from 23.4~62.4 Mb to 3.7~69.7 Mb and a peak falling between 53 and 60 Mb. The LOD score was bigger for non-fasting lipids than for fasting lipids: 6.78 for non-fasting non-HDL compared to 3.6 for fasting non-HDL, 5.33 for non-fasting triglyceride levels compared to 4.18 for fasting triglyceride levels. The LP allele elevated non-HDL cholesterol and triglyceride levels, while the BALB allele had an opposite effect on the traits. Details of this QTL and other QTL, including locus name, LOD score, peak location, 95% confidence interval (CI), high allele, mode of inheritance, and allelic effect, are presented in Table 1. This QTL was partially overlapping with *Hyplip2*, a locus for total, non-HDL cholesterol and triglycerides initially mapped in an intercross between MRL/MpJ and BALB/cJ mice and further validated in a congenic strain [33,34]. For triglycerides, this QTL replicates *Tglq3* mapped in a NZB/BlNJ × RF/J F2 cross [35].

A suggestive QTL on chromosome 3 was mapped for fasting non-HDL cholesterol levels (Figure 3A). This QTL had a suggestive LOD score of 2.61 and peaked at 129.5 Mb (Table 1). The LP allele was associated with higher non-HDL levels. This QTL replicates *Cq3*, mapped in B6 × KK-Ay F2 mice [36,37].

We identified a suggestive QTL on Chr2 for fasting HDL (Figure 3E) and one on Chr6 for non-fasting HDL cholesterol levels (Figure 3F). The Chr2 QTL had a suggestive LOD score of 2.62 and peaked at 26.4 Mb. This QTL was partially overlapping with *Hdl8* mapped in B6 × BALBc/ByJ *LDLr*^-/-^ F2 mice [38]. The Chr6 QTL had a suggestive LOD score of 3.18 and peaked at 138.6 Mb. This QTL replicates *Hdl6* mapped in B6 × DBA/2J F2 mice [39].

### 2.3. Coincident QTL for Plasma Lipids and Plasma Glucose

Interval mapping was conducted using MapManager QTX to define the 95% confidence interval of the QTL on Chr15 for non-HDL, triglycerides, and glucose (Figure 4). The yellow histograms underneath the peak of the LOD score curve denote the confidence interval of the QTL. The QTL for fasting non-HDL, *Nhdlq18*, had a confidence interval extending from the SNP marker gUNC25220278 (25.7 Mb) to mUNC25399622 (38.4 Mb) (Figure 4A), and the QTL for fasting triglycerides spanned a chromosomal region from the SNP marker gUNC25294263 (30.6 Mb) to mUNC150350596 (35.1 Mb) (Figure 4B). The lipid QTL was coincident with *Bglu20*, a significant QTL for fasting glucose, which extended from the SNP marker gUNC25294263 (30.6 Mb) to mUNC150350596 (35.1 Mb) (Figure 4C).

For non-fasting samples, the non-HDL QTL (*Nhdlq18*) had a confidence interval extending from the SNP marker S1L150578757 (14.5 Mb) to SBC151384683 (34.6 Mb) (Figure 4D), and the triglyceride QTL (*Tglq3*) extended from the marker mbHkupUNC150067245 (26.2 Mb) to gUNC25385155 (37.7 Mb) (Figure 4E). The lipid QTL was coincident with the QTL for plasma glucose (*Bglu20*), which extended from the SNP marker S1H150577925 (14.4 Mb) to S3N151341277 (33.5 Mb) (Figure 4F).

### 2.4. Correlations between Plasma Glucose and Lipid Levels

Under both fasting and non-fasting states, plasma glucose levels were closely correlated with plasma non-HDL cholesterol (fasting: r = 0.559; *p* = 3.83E−13; non-fasting: r = 0.650; *p* = 9.0E−19) and triglyceride levels (fasting: r = 0.654; *p* = 8.52E−19; non-fasting: r = 0.720; *p* = 1.85E−24) and slightly correlated with HDL cholesterol levels (fasting: r = 0.218; *p* = 9E−3; non-fasting: r = 0.257; *p* = 1.8E−3) among the F2 mice fed the Western diet (Figure 5A–F). Plasma triglyceride levels were more closely correlated with glucose levels than non-HDL cholesterol levels under either fasting or non-fasting states. Compared with the fasting samples, the non-fasting samples obtained from the same F2 mice showed closer correlations between plasma glucose and lipid levels.

On the chow diet, non-fasting plasma glucose levels showed modest correlations with non-HDL cholesterol (r = 0.431; *p* = 6.8E−8) and triglyceride levels (r = 0.47; *p* = 2.8E−9) but no correlation with HDL cholesterol levels (r = 0.022; *p* = 0.8) (Figure 5G–I).

### 2.5. Causal Relationship between Plasma Glucose and Lipids

Because the QTL for plasma glucose level was overlapping with the QTL for plasma non-HDL cholesterol and triglyceride levels on Chr15, and the plasma glucose levels were closely correlated with plasma lipid levels, we examined probable causal relationships between the traits. Residuals yielded from linear regression analysis of plasma glucose levels with plasma non-HDL cholesterol or triglyceride levels in the F2 mice were subjected to QTL mapping as a new phenotype. When the residuals from the regression analysis with fasting non-HDL cholesterol or triglyceride levels were analyzed, the significant QTL on Chr15 for fasting glucose disappeared (Figure 6). When residuals from regression analysis with fasting HDL cholesterol levels were analyzed, the Chr15 QTL for fasting glucose remained unaltered.

Similarly, when residuals from the regression analysis with non-fasting non-HDL cholesterol or triglyceride levels were analyzed, the significant QTL on Chr15 for non-fasting glucose disappeared (Figure 7). When residuals from the regression analysis with non-fasting HDL cholesterol levels were analyzed, the Chr15 QTL for non-fasting glucose showed little change. In addition, a suggestive QTL on Chr3 for plasma glucose levels showed up after eliminating the influence from fasting or non-fasting non-HDL cholesterol or triglycerides. This QTL peaked near 27 Mb and had a suggestive LOD score of 2.6~3.6.

### 2.6. Prioritization of Candidate Genes

QTL for plasma cholesterol and/or triglyceride levels on Chr15 have also been mapped in MRL/lpr × BALB/cJ [33], 129S1/SvImJ × CAST/Ei [40], NZB/BINJ × SM/J [41], and NZB/B1NJ × RF/J F2 mice [35]. At the QTL, the MRL/lpr, CAST/Ei, LP, and RF/J alleles raised lipid levels, while the BALB, 129S1/SvImJ, and NZB/B1NJ alleles had an opposite effect. Eleven genes within the confidence interval of *Nhdlq18* (from 23 to 69 Mb) contained one or more missense SNPs or SNP(s) in the upstream regulatory regions between the parental strains whose cross yielded the QTL (Table 2). These genes include *Oxr1*, *Tmem74*, *Samd12*, *Tnfrsf11b*, *Colec10*, *Deptor*, *Slc22a22*, *Zhx2*, *Mtss1*, *Sqle*, and *Nsmce2*. Of them, *Oxr1* and *Tmem74* contained one or more missense variants with a low SIFT score predicted to affect protein function.

## 3. Discussion

It is unclear if dyslipidemia is pathogenic to the development type 2 diabetes or merely a component of it. We sought to address this important problem using genetic means to determine the relationship of dyslipidemia with hyperglycemia, the defining measure of diabetes. The present study provides strong genetic evidence that non-HDL cholesterol and triglycerides play a causal role in the development of type 2 diabetes. We mapped a major QTL on chromosome 15 for fasting and non-fasting non-HDL cholesterol and triglyceride levels, which was colocalized with a major QTL for fasting and non-fasting plasma glucose levels. The QTL for plasma glucose disappeared after eliminating the influence from plasma non-HDL cholesterol or triglycerides. Moreover, plasma non-HDL cholesterol and triglyceride levels were closely correlated with plasma glucose levels in F2 mice under either fasting or non-fasting states in the F2 cohort.

Previous studies have provided conflicting results concerning the relationship between circulating lipids and type 2 diabetes. The latest meta-analysis of 33 randomized controlled trials including 21 with statins and 12 with proprotein convertase subtilisin/kexin type 9 (PCSK9) inhibitors shows that statins but not PCSK9 inhibitors are associated with the risk of incident diabetes [42], suggesting that the action of statins on diabetes is independent of reductions in non-HDL cholesterol levels. However, the longitudinal Framingham Heart Study indicates that low non-HDL cholesterol concentrations are associated with increased diabetes risk among individuals not treated with lipid-lowering drugs [28]. On the contrary, we observed a positive correlation between non-HDL cholesterol and glucose levels in a segregating F2 population on either a chow or Western diet. The correlation between the two traits was modest when the F2 mice developed moderate dyslipidemia on the chow diet and became stronger when mice developed severe dyslipidemia on the Western diet. Positive correlations between non-HDL cholesterol and glucose levels have been observed in other mouse crosses [23,24] and also in humans [43,44]. Compared to non-HDL cholesterol, plasma triglycerides were more closely associated with glucose levels in the F2 mice. Similar results have also been observed in humans [45]. Like non-HDL cholesterol, the triglycerides showed a stronger correlation with plasma glucose levels when the F2 mice were fed the Western diet relative to the chow diet. These observations support a more prominent role for triglycerides in hyperglycemia and diabetes development than non-HDL cholesterol.

An interesting finding of this study is the colocalization of the QTL for non-HDL cholesterol (*Nhdlq18*) and triglycerides with the QTL for plasma glucose (*Bglu20*) on chromosome 15. This colocalization allowed for elucidating the likely causal relationship between circulating lipids and diabetes. Using a causal inference test by excluding the influence from variation in plasma non-HDL cholesterol and triglycerides, we demonstrated the dependence of the glucose QTL on circulating lipids. Indeed, after adjustment for non-HDL cholesterol or triglycerides, the Chr15 QTL for fasting and non-fasting glucose disappeared. As fasting and non-fasting hyperglycemias are the defining features of diabetes, the current finding indicates that circulating lipids play a causal role in hyperglycemia and type 2 diabetes. Several factors contribute to the strength of our findings: First, the genetic connection to plasma glucose was found for both non-HDL cholesterol and triglycerides and also under both fasting and non-fasting states. Second, the LP allele, which raised plasma non-HDL cholesterol and triglyceride levels, was associated with elevated plasma glucose levels at the locus, while the BALB allele, which lowered lipid levels, was associated with reduced glucose levels. Finally, the power of linkage for the lipid QTL on chromosome 15 was parallel with the closeness of the correlation between lipids and glucose. Indeed, non-HDL cholesterol and triglycerides showed a closer correlation with glucose under the non-fasting state compared to the fasting state, and so the LOD scores were bigger for the lipids under this state.

We observed a weak positive correlation between HDL cholesterol and glucose levels in F2 mice on the Western diet, but no correlation in those on the chow diet. A weak positive correlation between the two traits had been observed in male F2 mice derived from an intercross between BALB and SM/J *Apoe^-/-^* strains [23], although a moderate inverse correlation was found in female F2 mice from the same intercross [24]. HDL is considered protective against type 2 diabetes due to its functions in cholesterol efflux, reverse cholesterol transport, anti-oxidation, anti-inflammation, and activation of the AMP-activated protein kinase [46,47]. Its positive correlation with plasma glucose levels suggests that HDL lost its anti-diabetic function when the F2 mice developed severe dyslipidemia on the Western diet. Indeed, a high fat diet and subsequent dyslipidemia induce oxidative stress and inflammation [48,49], which can alter the structure and function of HDL, making it dysfunctional [50].

As seen in humans [51,52], fasting and non-fasting levels of either non-HDL or HDL cholesterol were similar in the F2 mice. However, the triglyceride levels of the F2 mice showed a modest rise under the fasting state. Elevations in fasting triglyceride have also been observed in humans [53] and wild-type mice fed 2 weeks of atherogenic diets containing different amounts of fat and cholesterol [54]. A reduced clearance of very low density lipoproteins (VLDL), increased hepatic VLDL production, and reduced insulin release and sensitivity under the fasting state may contribute to elevations in fasting triglycerides [53,54].

Despite similar plasma levels, non-HDL cholesterol and triglycerides showed a closer correlation with glucose levels under the non-fasting state compared to the fasting state. This result is consistent with the larger LOD scores of the lipid QTL for non-fasting than fasting non-HDL cholesterol and triglycerides. Non-fasting lipid levels predict risk for coronary heart disease and stroke better than fasting lipid levels [55,56]. Our current study suggests that this is partially due to the genetic connection hyperglycemia and type 2 diabetes, which contributes to plaque growth [57].

Besides the current cross, the QTL for plasma non-HDL cholesterol and triglycerides on chromosome 15 have been mapped in the MRL/MpJ × BALB, NZB/B1NJ × RF/J, and 129 × CAST/Ei intercrosses [33,35,40]. Using the mapping and sequence variant data, we prioritized 11 candidate genes, all of which contained one or more missense SNPs or SNP(s) in the upstream regions segregating between the high allele and low allele strains. As 97% of the genetic variants between common mouse strains are ancestral [58], the QTL genes are almost certainly those containing polymorphisms shared among mouse strains. *Tnfrsf11b*, *Deptor*, and *Zhx2* are also functional candidates that may affect lipid or glucose metabolism.

In conclusion, the present study is the first to demonstrate the causal relationship of non-HDL cholesterol and triglyceride levels with type 2 diabetes in a cohort of F2 mice that developed dyslipidemia and hyperglycemia. Plasma non-HDL cholesterol and triglyceride levels were closely correlated with plasma glucose levels under both fasting and non-fasting conditions. Importantly, the QTL for non-HDL cholesterol and triglyceride levels was coincident with the QTL for plasma glucose levels, which disappeared after adjustments for the lipid levels. These observations provide a compelling argument for non-HDL cholesterol and triglycerides to serve as primary targets for the prevention and treatment of type 2 diabetes. However, it is noteworthy that the plasma cholesterol levels of *Apoe*^-/-^ mice on the Western diet were much higher than those of humans, and even those with familiar hyperlipidemia. There are many other differences between humans and mice in lipoprotein metabolism. For example, HDL is the major carrier of circulating cholesterol in wild-type mice, while LDL is the major carrier of cholesterol in humans. The current F2 mice were of the *Apoe*^-/-^ background and had a low fasting HDL cholesterol (21.4 ± 17.1 mg/dL) on the Western diet. Thus, further studies are needed to establish the causal relationship of non-HDL cholesterol and triglycerides with type 2 diabetes in humans.

## 4. Materials and Methods

### 4.1. Mice

BALB/cJ (BALB) and LP/J (LP) *Apoe^-/-^* mice were generated in our lab as reported [19]. LP-*Apoe^-/-^* males were crossed with BALB-*Apoe^-/-^* females to generate F1s, which were intercrossed to generate an F2 population. Mice were weaned at 3 weeks of age onto a regular chow diet, and at 6 weeks of age, started with a Western diet containing 21% fat, 34.1% sucrose, 0.15% cholesterol, and 19.5% casein (TD 88137, Envigo, Cumberland, VA, USA). Non-fasting blood was collected before and after 11 weeks on the Western diet. Fasting blood was collected after 12 weeks of the Western diet. All blood samples were drawn from the retro-orbital veins with the animals being anesthetized by isoflurane inhalation and prepared as reported [25]. All procedures were performed in accordance with the National Institutes of Health guidelines and approved on 20 January 2022 by the Institutional Animal Care and Use Committee (protocol #: 3109).

### 4.2. Glucose Assay

Plasma glucose concentrations were measured using a Sigma assay kit (Cat. # GAHK20, Saint Louis, MO, USA) as reported [25]. The plasma samples collected from the Western diet-fed mice were diluted 3× in H_2_O, and those from the chow diet-fed mice were not diluted. A total of 12 µL of samples, standards, and controls were loaded in 96-well plates and incubated with 150 µL per well of assay reagent for 30 min at 30 °C. Absorbance at 340 nm was measured with a Molecular Devices plate reader.

### 4.3. Lipid Assay

The plasma concentrations of total and HDL cholesterol and triglycerides were measured using reagents from Stanbio Labs (Boeme, TX, USA) and HDL precipitating reagent from Wako Diagnostics (Mountain View, CA, USA) as reported [25]. For the assay of total cholesterol concentrations in the Western diet-fed mice, plasma samples were diluted 3× with H_2_O. Non-HDL cholesterol was calculated as the difference between the total and HDL cholesterol levels.

### 4.4. Genotyping

DNA was prepared from the tails of mice with QIAGEN DNeasy Tissue Kits (San Diego, CA, USA). The resulting DNA samples were assessed for their quality and quantity using the ratio of absorbance at 260 and 280 nm. The average 260/280 ratio was 1.7 ± 0.1 and concentration was 58.3 ± 5.0 ng/µL (Appendix A). DNA samples were diluted to the final concentration of 20 ng/µL. All DNA samples submitted met the criteria of the genotyping company (Neogen). The F2 mice were genotyped at Neogen (Lansing, MI, USA) using miniMUGA arrays containing 11,000 SNP probes. The average call rate was 95% (Appendix A). Parental and F1 DNA served as controls on each array. Uninformative markers were excluded from QTL analysis. SNP markers that showed unexpected genotyping results for control samples or obvious departures from Hardy-Weinberg proportions were excluded. In addition, genotyping errors were checked using the “calc errorlod” function of R/qtl software. After filtration, 2595 SNP markers remained and were used for QTL mapping.

### 4.5. QTL Analysis

QTL mapping was performed using R/qtl and Map Manager QTX software as reported [48,59,60]. The thresholds for significant (*p* < 0.05) and suggestive (*p* < 0.63) LOD scores were based on 1000 permutations of the observed data for the autosomes and the X chromosome, the standard widely used in the QTL analysis of experimental crosses [61,62]. SA total of 1000 permutations were run at a 1-Mb interval across the entire genome for each trait. The allelic effect for a QTL was expressed as phenotype means ± SD for all three genotypes at its peak marker.

Interval mapping graphs for suggestive and significant QTL were constructed with QTX. For this function, redundant markers were excluded so that the remaining markers had distinct genotypes for the F2 cohort.

### 4.6. Evaluating Causal Relationships between Traits Based on Overlapping QTL

When the confidence intervals of QTL for two traits were overlapping, additional analysis was performed to examine potential causal relationships between them as described [48,63]. Briefly, residuals were generated from the linear regression analysis of two related traits and then subjected to genome-wide QTL mapping with the same algorithm previously used to map the overlapped QTL. The QTL yielded from the residuals should have been independent of one another.

### 4.7. Prioritization of Candidate Genes

Bioinformatics resources were used to prioritize candidate genes for lipid QTL that were mapped in two or more crosses derived from different parental strains. Variants were queried for via the Sanger Mouse Genomes Project (https://www.sanger.ac.uk/sanger/Mouse_SnpViewer/rel-1505; 28 March 2022). Potential candidate genes were those containing one or more missense SNPs or an SNP in an upstream regulatory region that segregated between high and low alleles as described [64,65,66]. A SIFT (Sorting Intolerant From Tolerant) score was obtained through Ensembl Genome Browser (https://useast.ensembl.org/index.html; 28 March 2022) and used for predicting the effect of a missense variant on protein function [67].

## Figures and Tables

**Figure 1 ijms-23-06184-f001:**
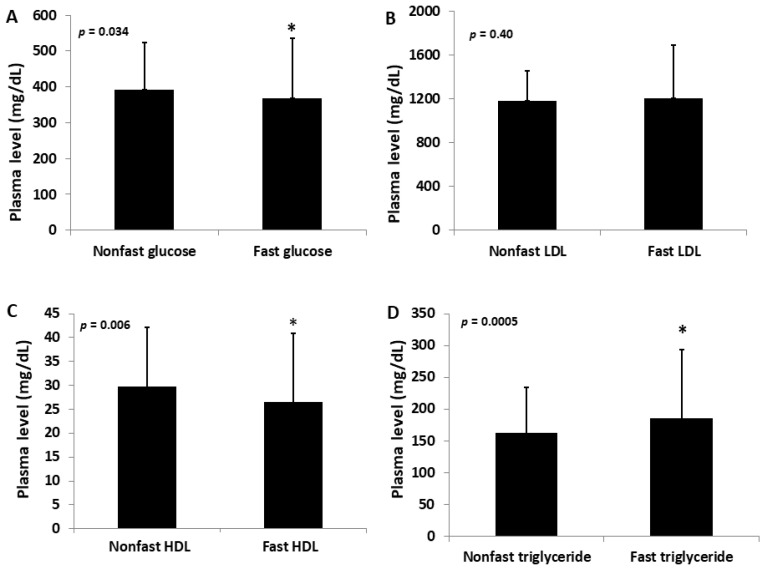
Fasting and non-fasting plasma levels of glucose (**A**), non-HDL (LDL) (**B**), HDL cholesterol (**C**), and triglycerides (**D**) in male F2 mice fed the Western diet. The results are expressed as means ± SD. * *p* < 0.05 vs. non-fasting measure.

**Figure 2 ijms-23-06184-f002:**
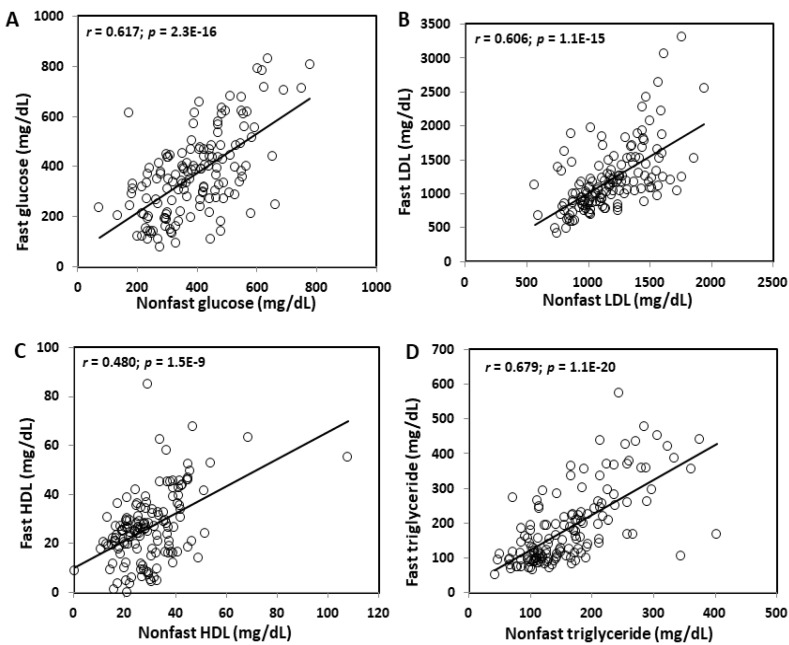
Correlations between fasting and non-fasting plasma levels of glucose (**A**), LDL (**B**), HDL cholesterol (**C**), and triglycerides (**D**) in the same F2 mice fed the Western diet. Each circle represents the values of an individual mouse. The correlation coefficient (r) and significance (*p*) are shown in the figures.

**Figure 3 ijms-23-06184-f003:**
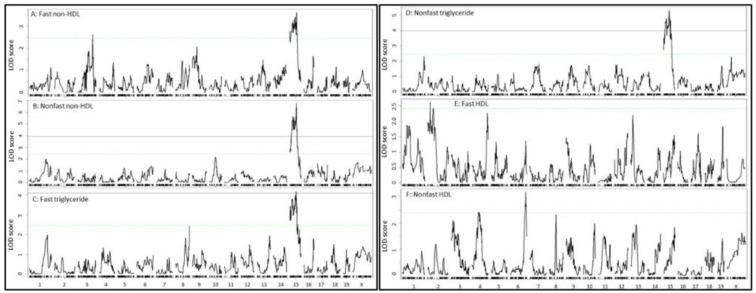
Genome-wide scans for loci influencing the plasma levels of fasting, non-fasting LDL (**A**,**B**), triglycerides (**C**,**D**), and HDL cholesterol (**E**,**F**) in male F2 mice fed the Western diet. Plots were created using the plotting function of R/qtl. Chromosomes 1 through X are represented on the x-axis. Each short vertical bar on the x-axis represents an SNP marker. The y-axis represents the LOD score. The two horizontal lines represent the genome-wide thresholds for significant (black) and suggestive linkage (green). Note the QTL on chromosome 15 for fasting plasma levels of non-HDL and triglycerides, but not HDL.

**Figure 4 ijms-23-06184-f004:**
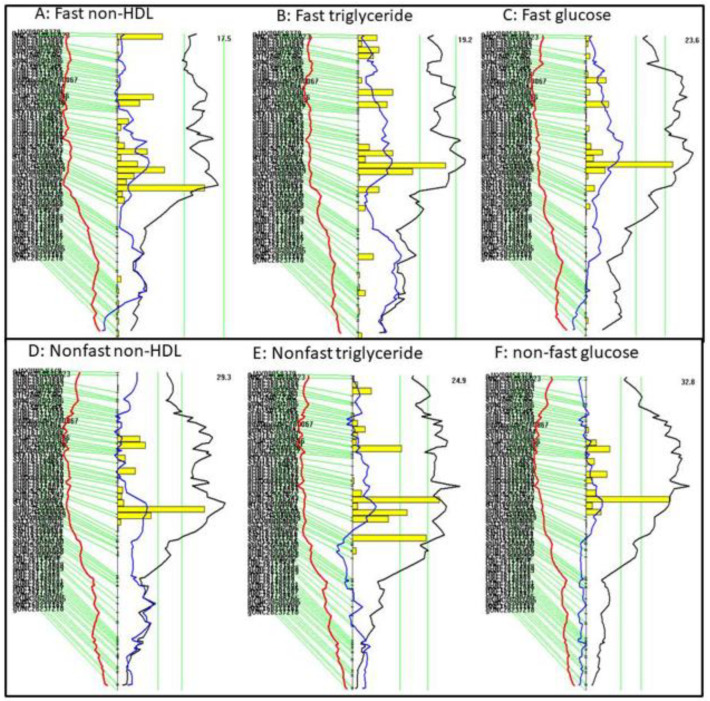
Interval mapping for QTL on chromosome 15 for the fasting plasma levels of LDL (**A**), triglycerides (**B**), and glucose (**C**), and the non-fasting plasma levels of LDL (**D**), triglycerides (**E**), and glucose (**F**). Plots were created using the interval mapping function of Map Manager QTX. The curved black line denotes the LOD score calculated at a 1 Mb interval along the chromosome. The red and blue lines denote additive and dominant regression coefficients, respectively. The yellow histograms denote confidence intervals estimated through bootstrap testing. The two vertical green lines denote genome-wide significance thresholds at *p* = 0.63 and *p* = 0.05, respectively. Genetic markers typed are shown on the left of each figure.

**Figure 5 ijms-23-06184-f005:**
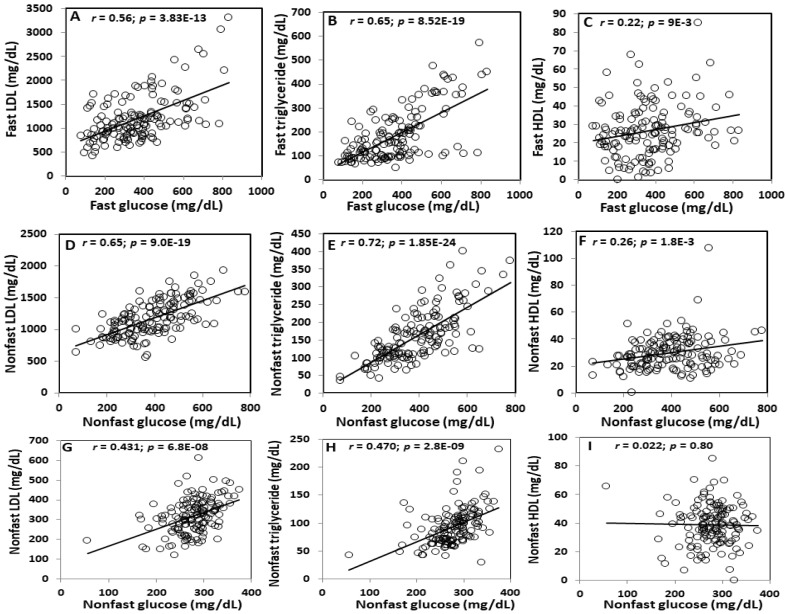
Correlations between plasma glucose and lipid levels in the same F2 mice fed the Western diet under fasting (**A**–**C**) and non-fasting states (**D**–**F**) and the chow diet under a non-fasting state (**G**–**I**). Each circle represents the values of an individual F2 mouse. The correlation coefficient (r) and significance (*p*) are shown in the figures.

**Figure 6 ijms-23-06184-f006:**
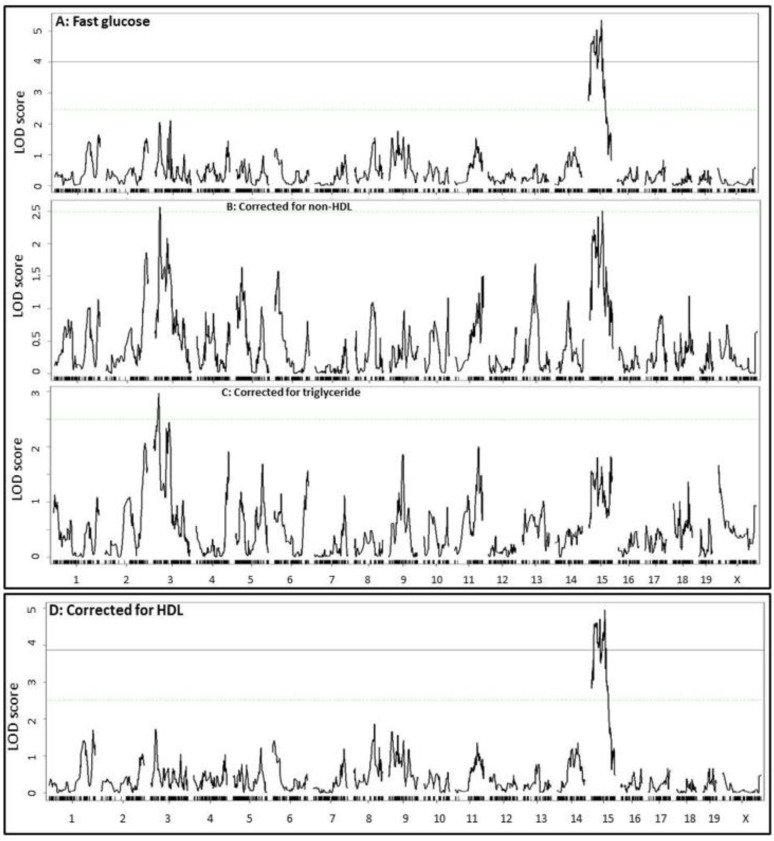
Genome-wide scans to assess the dependence of the QTL for fasting plasma glucose (**A**) on fasting plasma non-HDL (**B**), triglycerides (**C**), and HDL cholesterol levels (**D**) in the F2 mice. Residuals from the linear regression analysis of fasting glucose with fasting non-HDL, triglycerides, or HDL cholesterol levels were subjected to genome-wide scans to assess dependence on each other. Note the suppression or disappearance of the Chr15 QTL for fasting glucose after correction for non-HDL and triglycerides, but not HDL.

**Figure 7 ijms-23-06184-f007:**
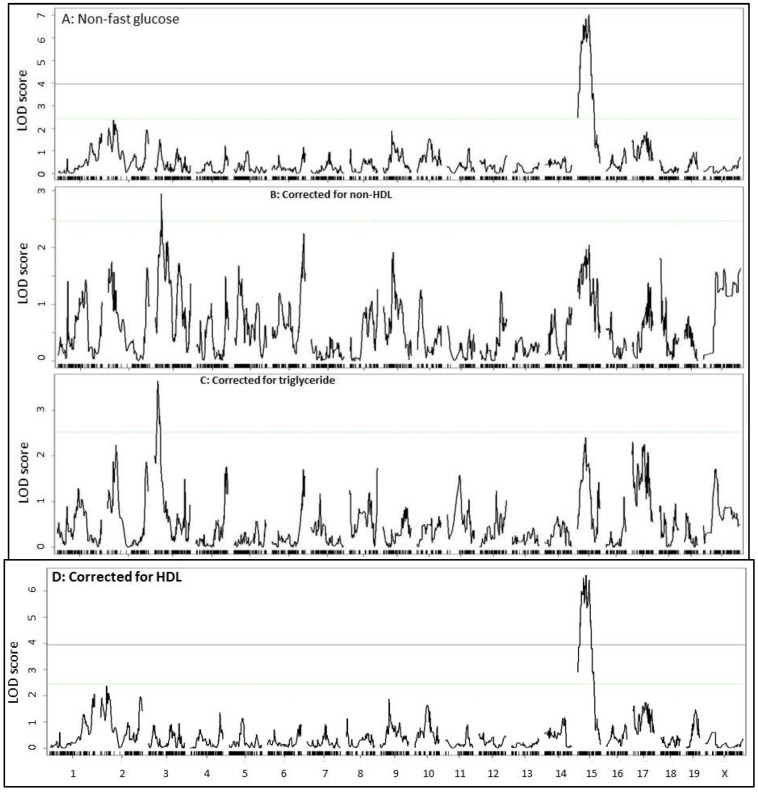
Genome-wide scans to assess the dependence of QTL for non-fasting glucose (**A**) on non-fasting non-HDL (**B**), triglycerides (**C**), and HDL cholesterol levels (**D**) in F2 mice. Residuals from the linear regression analysis of non-fasting glucose with non-fasting non-HDL, triglycerides, or HDL cholesterol levels were scanned. Note the disappearance of the Chr15 QTL for non-fasting glucose after correction for non-HD and triglycerides, but not HDL.

**Table 1 ijms-23-06184-t001:** Significant and suggestive QT for lipid levels mapped with male F2 mice derived from LP and BALB *Apoe*^−/−^ mice.

Locus Name	Chr	LOD ^a^	Peak (Mb)	Closest Marker	95%CI (Mb) ^b^	High Allele	Mode of Inheritance	Allelic Effect ^c^		
								BB	H	LL
Non-HDL (non-fast)										
*Nhdlq18*, *Hyplip2*	15	**6.78**	59.7	gUNC25683390	23.4–62.4	LL	Additive	981 ± 188	1217 ± 269	1284 ± 273
Non-HDL (fast)										
*Nhdlq18*, *Hyplip2*	15	3.60	64.7	mJAX00062941	3.7–69.7	LL	Additive	953 ± 364	1211 ± 443	1383 ± 537
*Cq3*	3	2.61	129.5	UNC6226186	46.8–152.8	LL	Recessive	1298 ± 476	1066 ± 333	1388 ± 629
HDL (fast)										
	2	2.62	26.4	gUNC2767893	4.4–76.4	-	Heterosis	33.2 ± 11.3	26.5 ± 9.4	35.7 ± 16.5
HDL (non-fast)										
	6	3.18	138.6	S6R065543032	126.7–145.7	LL	Recessive	30.8 ± 11.3	26.5 ± 9.4	35.7 ± 16.5
Triglyceride (non-fast)										
*Tglq3*	15	**5.33**	53.2	gUNC25604126	16.1–66.3	LL	Additive	112 ± 38	165 ± 68	192 ± 80
Triglyceride (fast)										
*Tglq3*	15	**4.18**	59.0	gUNC25683390	5.7–64.9	LL	Additive	122 ± 55	202 ± 109	216 ± 118

^a^ LOD scores were obtained from genome-wide QTL analysis using R/qtl. Significant LOD scores are highlighted in bold. ^b^ 95% Confidence intervals are in Mb for significant or suggestive QTL. ^c^ BB: BALB allele; LL: LP allele; H: heterozygous for both BALB and LP alleles. Unit for plasma lipids: mg/dL. Values are expressed as means ± SD.

**Table 2 ijms-23-06184-t002:** Candidate genes for Nhdlq18 on chromosome 15 identified by haplotype analysis.

Chr	Position	Gene	dbSNP	Ref	LP_J	RF_J	CAST_EiJ	BALB_cJ	129P2_OlaHsd	NZB_B1NJ	Csq	AA	AA Coord	SIF T
15	3979418	Fbxo4	rs263501543	C	T *	-	-	-	T *	T *	missense_variant	G/R	3	0.13
15	3980692	Fbxo4	rs31620120	G	A *	-	-	-	A *	A *	upstream_gene_variant			
15	4370841	Plcxd3	rs31896942	G	A	-	-	-	A	A	upstream_gene_variant			
15	4371744	Plcxd3	rs32133183	A	G	-	-	-	G	G	upstream_gene_variant			
15	4371814	Plcxd3	rs32535204	G	A	-	-	-	A	A	upstream_gene_variant			
15	4373929	Plcxd3	rs31966655	A	G	-	-	-	G	G	upstream_gene_variant			
15	25413871	Basp1	rs262340099	C	T	-	-	-	t	T	upstream_gene_variant			
15	32173458	Tas2r119	rs51514737	A	T	-	-	-	T	-	upstream_gene_variant			
15	32173469	Tas2r119	rs46216365	A	G	-	-	-	G	-	upstream_gene_variant			
15	41819897	Oxr1	rs50179186	T	G	-	G	-	G	G	missense_variant	S/A	239	0.02
15	41820278	Oxr1	rs31574788	A	C	-	C	-	C	C	missense_variant	T/P	428	0.2
15	41820327	Oxr1	rs31850612	C	T	-	-	-	T	T	missense_variant	S/L	363	0.15
15	43283531	Eif3e	rs49218373	A	G	-	-	-	G	-	upstream_gene_variant			
15	43285477	Eif3e	rs46145200	G	A	-	-	-	A	-	upstream_gene_variant			
15	43285631	Eif3e	rs52016538	G	A	-	-	-	A	-	upstream_gene_variant			
15	43286989	Eif3e	rs52120197	T	C	-	-	-	C	-	upstream_gene_variant			
15	43287113	Eif3e	rs578378356	T	C	-	-	-	C	-	upstream_gene_variant			
15	43287574	Eif3e	rs585572417	A	T	-	-	-	T	-	upstream_gene_variant			
15	43870636	Tmem74	rs32033816	G	T	-	-	-	T	-	upstream_gene_variant			
15	43870825	Tmem74	rs32094059	A	T	-	-	-	T	-	upstream_gene_variant			
15	43872938	Tmem74	rs52337392	A	T	-	-	-	T	-	upstream_gene_variant			
15	43873253	Tmem74	rs31706856	T	C	-	-	-	C	-	upstream_gene_variant			
15	44191397	Trhr	rs52202529	T	C	-	-	-	C	-	upstream_gene_variant			
15	44194654	Trhr	rs32144218	A	C	-	-	-	C	C	upstream_gene_variant			
15	44194999	Trhr	rs32516067	G	A	-	-	-	A	-	upstream_gene_variant			
15	44195817	Trhr	rs31913481	A	G	-	-	-	G	-	upstream_gene_variant			
15	44196286	Trhr	rs33306588	C	T	-	-	-	T	-	5_prime_utr_variant			
15	44400055	Nudcd1	rs51311008	G	A	-	-	-	A	-	synonymous_variant			
15	44405503	Nudcd1	rs32045629	G	A	-	-	-	A	-	synonymous_variant			
15	44452953	Pkhd1l1	rs3716450	A	G	-	-	-	G	-	upstream_gene_variant			
15	44453339	Pkhd1l1	rs32007056	G	A	-	-	-	A	-	upstream_gene_variant			
15	44453354	Pkhd1l1	rs31937456	G	A	-	-	-	A	-	upstream_gene_variant			
15	44453478	Pkhd1l1	rs31864580	A	G	-	-	-	G	-	upstream_gene_variant			
15	44453784	Pkhd1l1	rs31916062	A	G	-	-	-	G	-	upstream_gene_variant			
15	44454507	Pkhd1l1	rs31571652	G	T	-	-	-	T	-	upstream_gene_variant			
15	44454738	Pkhd1l1	rs31772927	G	T	-	-	-	T	-	upstream_gene_variant			
15	44456205	Pkhd1l1	rs50758342	A	G	-	-	-	G	-	upstream_gene_variant			
15	44457111	Pkhd1l1	rs31671052	C	T	-	-	-	T	-	upstream_gene_variant			
15	44457216	Pkhd1l1	rs31628733	T	C	-	-	-	C	-	upstream_gene_variant			
15	44457499	Pkhd1l1	rs33288852	G	A	-	-	-	A	-	upstream_gene_variant			
15	44457534	Pkhd1l1	rs239874708	C	G	-	-	-	G	-	upstream_gene_variant			
15	44457535	Pkhd1l1	rs259180475	C	T	-	-	-	T	-	upstream_gene_variant			
15	44457539	Pkhd1l1	rs217867183	G	A	-	-	-	A	-	upstream_gene_variant			
15	44493144	Pkhd1l1	rs33290607	C	T	-	-	-	T	-	missense_variant	T/I	** 335 **	**0.22**
15	44586472	Pkhd1l1	rs32502839	A	C	-	C	-	C	-	missense_variant	N/T	3877	**0.01**
15	44746395	Sybu	rs31952285	C	G *	-	-	-	G *	-	missense_variant	R/P	159	**0**
15	44787720	Sybu	rs32209513	A	G *	-	G *	-	G *	-	missense_variant	F/L	63	0.9
15	45109081	Kcnv1	rs13482546	C	T	-	-	-	T	-	missense_variant	V/I	469	0.95
15	45114557	Kcnv1	rs51296409	G	A	-	-	-	A	-	synonymous_variant			
15	45115527	Kcnv1	rs32212679	G	T	-	-	-	T	T	upstream_gene_variant			
15	45115974	Kcnv1	rs32021420	C	A	-	-	-	A	-	upstream_gene_variant			
15	45116156	Kcnv1	rs31825726	G	A	-	-	-	A	-	upstream_gene_variant			
15	45116355	Kcnv1	rs32046135	C	T	-	-	-	T	-	upstream_gene_variant			
15	45117030	Kcnv1	rs32376626	G	T	-	-	-	T	-	upstream_gene_variant			
15	45118047	Kcnv1	rs48752875	T	A		-	-	A	-	upstream_gene_variant			
15	45118200	Kcnv1	rs45851350	G	A	-	-	-	A	-	upstream_gene_variant			
15	45118590	Kcnv1	rs50814374	T	C	-	-	-	C	-	upstream_gene_variant			
15	45118760	Kcnv1	rs47234445	T	C	-	-	-	C	-	upstream_gene_variant			
15	54252267	Tnfrsf11b	rs31629761	C	T	-	-	-	T	-	synonymous_variant			
15	54252313	Tnfrsf11b	rs31799791	A	C	-	-	-	C	-	missense_variant	L/R	296	0.61
15	54256164	Tnfrsf11b	rs33484516	C	T	-	-	-	T	-	missense_variant	R/Q	138	0.23
15	54278337	Tnfrsf11b	rs31923434	G	A	-	-	-	A	-	5_prime_utr_variant			
15	54278530	Tnfrsf11b	rs47057076	G	T	-	-	-	T	-	upstream_gene_variant			
15	54278620	Tnfrsf11b	rs49814729	A	G	-	-	-	G	-	upstream_gene_variant			
15	54278628	Tnfrsf11b	rs33490015	A	G	-	-	-	G	-	upstream_gene_variant			
15	54278937	Tnfrsf11b	rs51583114	T	C	-	-	-	C	-	upstream_gene_variant			
15	54278947	Tnfrsf11b	rs33489239	A	G	-	-	-	G	-	upstream_gene_variant			
15	54278995	Tnfrsf11b	rs51682935	G	A	-	-	-	A	-	upstream_gene_variant			
15	54279003	Tnfrsf11b	rs46278323	C	T	-	-	-	T	-	upstream_gene_variant			
15	54279030	Tnfrsf11b	rs33489235	G	C	-	-	-	C	-	upstream_gene_variant			
15	54279073	Tnfrsf11b	rs33488583	T	C	-	-	-	C	-	upstream_gene_variant			
15	55180943	Deptor	rs48756035	C	T	-	-	-	T	T	synonymous_variant			
15	55220217	Deptor	rs32271813	G	A *	-	-	-	A *	A *	missense_variant			

Chr: chromosome; Position: in bp; dbSNP: Single nucleotide polymorphism database; Ref: Reference or C57BL/6J SNP; Csq: SNP consequences. AA: Amino acid; AA coord: Amino acid coordinate. SIFT, Sorting Intolerant From Tolerant (intolerant SNP is highlighted in bold). “-” same as reference SNP. Not all upstream variants were shown due to space limitation. * Multiple consequences.

## Data Availability

All data reported in this article are included in the Appendix A.

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
