# Peer review of "Genetic Evidence for a Causal Relationship between Hyperlipidemia and Type 2 Diabetes in Mice"

_ijms, 2022, doi:10.3390/ijms23116184_

Round 1

Reviewer 1 Report

Dear authors,

This work attempts to analyze the genetic relationship between hyperlipidemia and type 2 diabetes, using Apoe -/- generated mice. Although the results of this study could be very interesting, the manuscript has important errors that must be corrected before carrying out an in-depth review of both the results and the conclusions drawn. Specifically, the authors must add line numbers to simplify the review, they must justify the text, they must eliminate information about results in the introduction of the manuscript, the "Methods" section must be renamed as "Material and Methods", since it includes information about the material used, such as the animals used in the study.

In addition, the information in this section is scarce, since the authors have not indicated how the DNA extraction has been carried out, from what biological material, what quality and quantity results have been obtained, etc.

Other errors are, for example, indicating that a p-value <0.63 is "suggestive". What do the authors mean by this? In statistics, a result is significant or not, but I had never read that a result was "suggestive". I recommend that the authors review the manuscript in depth and improve both its presentation and the information provided.

Author Response

Comment: This work attempts to analyze the genetic relationship between hyperlipidemia and type 2 diabetes, using Apoe -/- generated mice. Although the results of this study could be very interesting, the manuscript has important errors that must be corrected before carrying out an in-depth review of both the results and the conclusions drawn. Specifically, the authors must add line numbers to simplify the review, they must justify the text, they must eliminate information about results in the introduction of the manuscript, the "Methods" section must be renamed as "Material and Methods", since it includes information about the material used, such as the animals used in the study.

Response: We have made the suggested revisions.

Comment: In addition, the information in this section is scarce, since the authors have not indicated how the DNA extraction has been carried out, from what biological material, what quality and quantity results have been obtained, etc.

Response: Amended.

Comment: Other errors are, for example, indicating that a p-value <0.63 is "suggestive". What do the authors mean by this? In statistics, a result is significant or not, but I had never read that a result was "suggestive". I recommend that the authors review the manuscript in depth and improve both its presentation and the information provided.

Response: Thresholds for significant (P < 0.05) and suggestive (P < 0.63) LOD scores were based on 1,000 permutations of the observed data for the autosomes and the X chromosome.   This standard is widely used in QTL analysis (K.W. Broman, S. Sen, S.E. Owens, A. Manichaikul, E.M. Southard-Smith, G.A. Churchill. The X chromosome in quantitative trait locus mapping Genetics., 174 (2006), pp. 2151-2158. https://www.sciencedirect.com/science/article/pii/S0022227520408582#bib29).

Reviewer 2 Report

It is a well written and well presented article that is easy to follow and adds to the knowledge in the area and the results point to a relationship between non-HDL cholesterol and triglyceride levels with type 2 diabetes, at lease in the cohort of mice that were examined. The article needs however to also acknowledge the possible differences between the animals and human subjects.

Author Response

Comment: It is a well written and well presented article that is easy to follow and adds to the knowledge in the area and the results point to a relationship between non-HDL cholesterol and triglyceride levels with type 2 diabetes, at lease in the cohort of mice that were examined. The article needs however to also acknowledge the possible differences between the animals and human subjects.

Response: We agree with the reviewer’s comment and have addressed the issue in Discussion.

Round 2

Reviewer 1 Report

the authors have not followed my format recommendations and have not indicated the quantity and quality of the extracted DNA

Author Response

Comment: the authors have not followed my format recommendations and have not indicated the quantity and quality of the extracted DNA

Response: The quantity and quality of the extracted DNA samples were evaluated using the absorbance at 260 nm and 280 nm in our lab.  The ratio of 260/280 in absorbance was 1.70. The quality of DNA samples was also evaluated by the genotyping company and was “excellent”.  The data are included in Supplemental materials.
